# A General Integrated Method for Design Analysis and Optimization of Missile Structure

**Xiaoguang Wang** [1] , **Jun Yang** [2] , **Jian Guo** [3] **and Jun Guo** [1,*]

1   School of Astronautics, Beijing University of Aeronautics and Astronautics, Beijing 100191, China; wangxiaoguang@buaa.edu.cn
2   Shanghai Aerospace Control Technology Institute (SACTI), Shanghai 200233, China; qyj0578_cn@126.com
3   Baicheng Ordnance Test Center of China, Baicheng 137001, China; guojianbc@163.com
*   Correspondence: guojunbh@buaa.edu.cn

**Abstract:** In the demonstration phase of a missile scheme, to obtain the optimum proposal, designers need to modify the parameters of the overall structure frequently and significantly, and perform the structural analysis repeatedly. In order to reduce the manual workload and improve the efficiency of research and development, a general integrated method of missile structure modeling, analysis and optimization was proposed. First, CST (Class and Shape transformation functions) parametric method was used to describe the general structure of the missile. The corresponding software geometric modeling and FEM (Finite Element Method) analyzing of the missile were developed in C/C++ language on the basis of the CST parametric method and UG (Unigraphics) secondary development technology. Subsequently, a novel surrogate model-based optimation strategy was proposed to obtain a relatively light mass missile structure under existing shape size. Eventually, different missile models were used to verify the validity of the method. After executing the structure modeling, analysis and optimization modules, satisfactory results can be obtained that demonstrated the stability and adaptability of the proposed method. The method presented saves plenty of time comparing to the traditional manual modeling and analysis method, which provides a valuable technique to improve the efficiency of research and development.

**Keywords:** missile modeling; structure analysis; parameterization; CST; UG secondary development; surrogate model; optimization strategy

## 1. Introduction

The research and manufacture of missile weapon system is a complicated systematic project, which involves many technical fields and departments. From the stage of design to mass production, it needs to undergo the processes of feasibility demonstration, program demonstration, preliminary design, sample design, design typification [1]. Generally speaking, the conceptual design of missile mainly relies on the personnel experience of engineering technicians, which is an iterative process that requires a serious amount of time and energy to obtain an optimal scheme, and the typical duration for a conceptual design activity is approximately a few months. Hence, realizing the rapid conceptual design of missiles will have extremely vital significance to improve the design efficiency of the missile, and shorten the production cycle.

UG is a high-end software product for engineering solutions that includes CAD/CAE/CAM modules. Meanwhile, the UG system provides powerful secondary tools of UG/OPEN API, Menu Script and UIStyler development to customize user interface and application programs as needed. Pegemanyfar [2] presented a design tool, which was for the combustion chamber preliminary design, driven by the EXCEL database and generated by the UG software. LIU et al. [3] developed an automated

3D modeling program based on the UG secondary development technology, which has improved the modeling efficiency. Wang et al. [4] used the technology of UG/KF secondary development for the automatic modeling of the wind turbine blades, and the aerodynamic characteristics of the impeller were analyzed by the Fluent software. Moreover, He et al. [5] implemented the parametric modeling of the blended-wing-body underwater glider structure by using UG secondary development technology and studied its structural performance with ANSYS. Therefore, UG is considered as an excellent tool that can realize the integration of modeling and simulation.

Meanwhile, airfoils have been parameterized in a wide variety of ways. Among them, CST is considered to be a more effective method, which can represent the different types of geometries in a generic way with fewer parameters. Kulfan and Bussoletti [6] proposed a parameterization method that based on class and shape transformation functions, which has the advantages of high precision and fewer variables. The CST method is a powerful and versatile means to describe complex aeronautical shapes ranging from 2D airfoils to 3D geometries of aircraft using analytical functions. Since it has been introduced, this parameterization method has been the subject of several studies [7]. Marco Ceze of the University of Michigan studied the characteristics of CST parameterization and found that the ill-conditioning of parameterization when fitting airfoil with high-order Bernstein polynomials [8]. Straathof et al. proposed a parameterization method that used a combination of Bernstein polynomials and B-splines to allow for both the local and global control of a shape, which is an extension to the Class-Shape-Transformation Method [9].Currently, for 3D wing geometries, the CST methodology is being widely used to drive aerodynamic optimization problems by using a full potential panel code, where the Euler solver being used for 2D airfoil optimizations [10–13].

Moreover, in terms of structural optimization, plenty of research has been made, and various optimization strategies have been proposed. Presently, the surrogate model technique is promising due to the excellent computational accuracy and efficiency, especially for the practical engineering system with complex mapping functions [14–17]. The existing surrogate models, for instance the Polynomial Regression, Radial Basis Functions, Kriging models, Support Vector Regression, and Polynomial chaos expansion, have their limitations. Therefore, the proposal of a high-efficiency and straightforward surrogate model will be beneficial for improving the efficiency and accuracy of computational response.

In this paper, UG secondary development technology combined with powerful CST parameterization methodology is used to develop an automatic geometric modeling and FEM structural analysis system for the conceptual missile design, and a novel surrogate model-based optimization strategy is designed to obtain a relatively light mass missile structure under the existing shape size. Moreover, different examples are provided to verify the feasibility of the proposed method for the rapid conceptual design of the missile structure, and this can significantly improve the design efficiency and shorten the design cycle of the missile.

## 2. Theory

### 2.1. CST Parameterization Method

The CST geometry representation method is a powerful parameterization that can efficiently model any airfoil in the entire design space. This method combines an analytical "class function" with a parametric "shape function," where the class function describes an essential class of shapes, and the shape function describes the permutation around this basic shape.

The CST general form of the mathematical expression representing the airfoil geometry is:

$$\frac{z}{c}\left(\frac{x}{c}\right) = \left(\frac{x}{c}\right)^{N_1} \cdot \left(1 - \frac{x}{c}\right)^{N_2} \cdot \sum_{i=0}^{n}\left[A_i\left(\frac{x}{c}\right)^i\right] + \frac{x}{c} \cdot \frac{z_{TE}}{c}, \tag{1}$$

where $c$ is chord length, the exponents $N_1$ and $N_2$ define the type of geometry, $\left(\frac{x}{c}\right)^{N_1}\cdot\left(1-\frac{x}{c}\right)^{N_2}$ is the general expression of class function, $\sum_{i=0}^{n}\left[A_i\cdot\left(\frac{x}{c}\right)^i\right]$ is a general function that describes the unique shape of the geometry between the nose and the aft end, and the term $\frac{z_{TE}}{c}$ provides trailing edge thickness.

Correspondingly, we define the term of $C\left(\frac{x}{c}\right)$ and $S\left(\frac{x}{c}\right)$ to simplify the form of expression:

$$\frac{z}{c}\left(\frac{x}{c}\right) = C\left(\frac{x}{c}\right)\cdot S\left(\frac{x}{c}\right) + \frac{x}{c}\cdot\frac{z_{TE}}{c}, \tag{2}$$

where

$$C\left(\frac{x}{c}\right) = \left(\frac{x}{c}\right)^{N_1}\cdot\left[1-\frac{x}{c}\right]^{N_2} \tag{3}$$

$$S\left(\frac{x}{c}\right) = \sum_{i=0}^{n}\left[A_i\cdot\left(\frac{x}{c}\right)^i\right]. \tag{4}$$

The CST parameterization method represents a two-dimensional geometry by the product of a class function $C\left(\frac{x}{c}\right)$, and a shape function $S\left(\frac{x}{c}\right)$, plus a term that characterizes the trailing edge thickness, as denoted in Equation (2).

In addition, the first and last term of the shape function is determined by imposing boundary conditions on the airfoil shape; the result directly related to the airfoil leading edge nose radius $R_{LE}$, trailing edge boattail angle $\beta$, and the thickness $z_{TE}$:

$$S(0) = \sqrt{\frac{2R_{LE}}{c}} \tag{5}$$

$$S(1) = \tan\beta + \frac{z_{TE}}{c}. \tag{6}$$

As mentioned, the class function coefficients $N_1$ and $N_2$ determine the basic shape, whereas the shape function provides the relation that transforms the airfoil physical domain into a well behaved analytic function. The shape function is represented by a Bernstein polynomial of arbitrary order n, which is composed of n+1 scalable components, and the individual components in the polynomial can be scaled to define a wide variety of airfoil geometries.

The definition of the order n Bernstein polynomial is:

$$BP_n = \sum_{r=0}^{n}[K_{r,n}\left(\frac{x}{c}\right)^r\cdot\left(1-\frac{x}{c}\right)^{n-r}]. \tag{7}$$

The shape function can be recast as:

$$S\left(\frac{x}{c}\right) = \sum_{r=0}^{n}\left[v_r\cdot S_{r,n}\left(\frac{x}{c}\right)\right], \tag{8}$$

where

$$S_{r,n}\left(\frac{x}{c}\right) = K_{r,n}\left(\frac{x}{c}\right)^r\cdot\left(1-\frac{x}{c}\right)^{n-r} \tag{9}$$

$$K_{r,n} = \binom{n}{r} = \frac{n!}{r!(n-r)!}. \tag{10}$$

Assume there exists a set of control points $[x_i, z(x_i)]$. Therefore, we can build a set of equations that relates geometry control points $[x_i, z(x_i)]$ directly to weighting coefficients $v_r$:

$$\begin{bmatrix} M_0(x_0) & M_1(x_0) & M_2(x_0) & \dots & M_n(x_0) \\ M_0(x_1) & M_1(x_1) & M_2(x_1) & \dots & M_n(x_1) \\ \vdots & \vdots & \vdots & \ddots & \vdots \\ M_0(x_n) & M_1(x_n) & M_2(x_n) & \dots & M_n(x_n) \end{bmatrix} \begin{Bmatrix} v_0 \\ v_1 \\ \vdots \\ v_n \end{Bmatrix} = \begin{Bmatrix} z(x_0) - x_0\cdot z_{TE} \\ z(x_1) - x_1\cdot z_{TE} \\ \vdots \\ z(x_n) - x_n\cdot z_{TE} \end{Bmatrix}, \tag{11}$$

where

$$M_r\left(\frac{x}{c}\right) = C\left(\frac{x}{c}\right) \cdot S_{r,n}\left(\frac{x}{c}\right). \tag{12}$$

Examples of the ability of the CST method to represent typical airfoils NACA0006 and NACA4412, and the residual differences between the actual airfoil and the approximated are shown in Figures 1 and 2. In the figure of the residual differences, the range of 0~1 and -1~0 represent the upper and lower airfoil surface, respectively. Note from the figures that the differences between the actual NACA airfoil and the approximated airfoil are hardly discernible even for the Bernstein polynomial of order 5. In general, the shape function with BPO = 6~9 can satisfy the requirement of high precision [18].

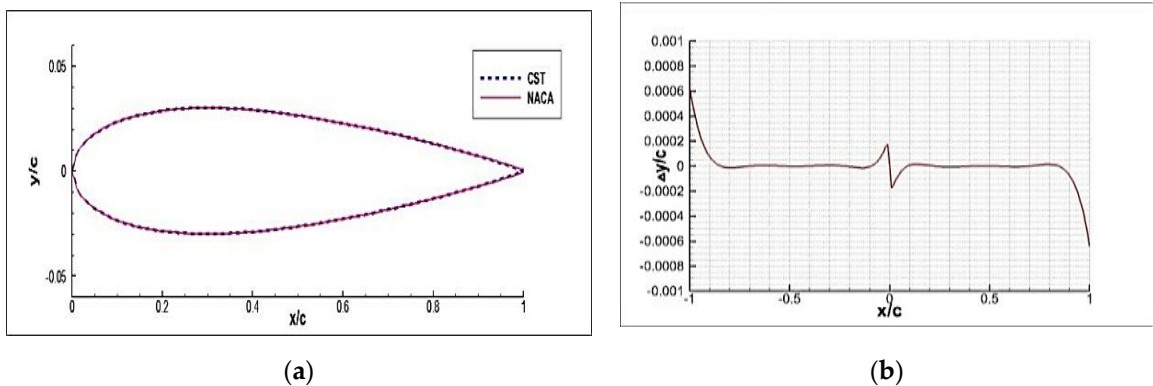

(**a**)  (**b**)

**Figure 1.** The CST parametrization of an NACA0006 airfoil and the corresponding fitting residual with BPO = 5: (**a**) The airfoil geometric shape and (**b**) the residual differences.

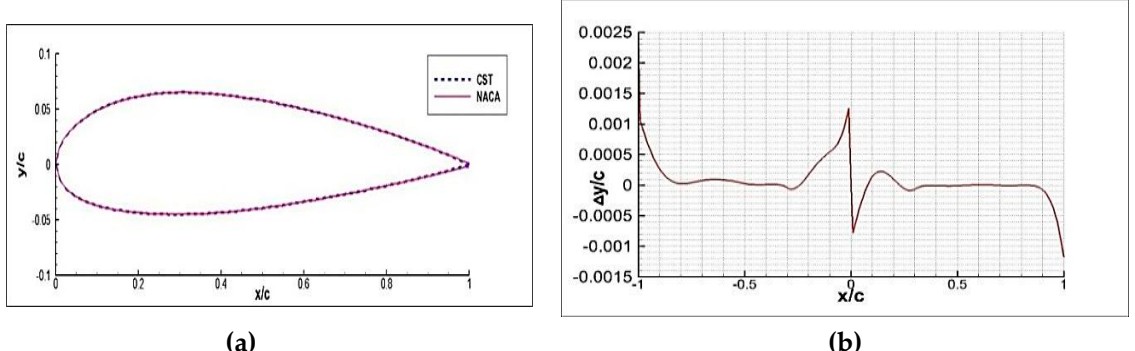

(**a**)  (**b**)

**Figure 2.** The CST parametrization of an NACA4412 airfoil and the corresponding fitting residual with BPO = 5: (**a**) The airfoil geometric shape and (**b**) the residual differences.

As previously shown, the CST parameterization methodology can efficiently model any airfoil in the entire design space, and it has the advantage of high precision and fewer variables. Hence, the CST method will be used to represent the wing and body shape of the missile, which can reduce the design variable number and improve the modeling accuracy in the entire missile design loops.

### 2.2. Augmented Fourier Series-Based Polynomial Surrogate Model

Generally speaking, the optimization process needs to call the parametric modeling module and analysis module several times, which directly obstructs the implementation of the optimization strategy due to the vast amount of computation. In recent years, the surrogate model has been proven to be a useful technique according to its excellent computational efficiency. The system response, approximated as a simple explicit function, can be easily derived without executing tedious computation. In this section, a novel Augmented Fourier series-based polynomial (AFP) surrogate model will be constructed to replace the original finite element calculation [19].

If a periodical function $f(x)$ is continuous over $[-\pi, \pi]$, then exists a Fourier series $F(x)$ which converges to the function $f(x)$ on $[-\pi, \pi]$ for any $\varepsilon > 0$ that is expressed as [20]:

$$\|f(x) - F(x)\|_\infty < \varepsilon , \quad x \in [-\pi, \pi] . \tag{13}$$

Let $F_n(x)$ denote the set of Fourier series of degree not larger than $n$. For any non-negative integer $n$, there exists a unique series $F_n^*(x)$ that:

$$\begin{aligned} \|f(x) - F_n(x)\|_\infty \quad &\geq \|f(x) - F_n^*(x)\|_\infty \\ &= \inf\|f(x) - F_n(x)\|_\infty < \varepsilon \end{aligned} \tag{14}$$

where $F_n^*(x)$ is the best uniform approximation of degree $n$ to $f(x)$ over $[-\pi, \pi]$. However, it is difficult to obtain the best uniform approximation Fourier series. The best square approximation Fourier series obtained by using 2-norm can be used to approximate the original function.

Then, the function $f(x)$ over the interval $[-\pi, \pi]$ can be approximated by the Fourier series of degree $n$ in the combined cosine and sine function forms:

$$f(x) \approx f_{F_n}(x) = \frac{a_0}{2} + \sum_{i=1}^{n} (a_i \cos ix + b_i \sin ix). \tag{15}$$

Without losing generality, we consider a continuous function $g(x)$ defined on an arbitrary interval $[\underline{x}, \overline{x}]$. Then, the Fourier polynomial of degree $n$ can be expressed as:

$$\begin{aligned} g(x) \approx g_{F_n}(x) \quad &= \tfrac{1}{2}g_0 + \sum_{i=1}^{2n} g_i T_i(x) \\ &= \tfrac{1}{2}g_0 + \sum_{i=1}^{2n} g_i T_i(\theta) , \quad \theta \in [-\pi, \pi] \end{aligned} \tag{16}$$

where

$$T_i(x) = T_i(\theta) = \begin{cases} 1 & i = 0 \\ \cos k\theta & i = 2k-1, \quad k = 1, 2, \cdots, n \\ \sin k\theta & i = 2k, \quad k = 1, 2, \cdots, n \end{cases}$$

$$x = \frac{\underline{x} + \overline{x}}{2} + \frac{\overline{x} - \underline{x}}{2\pi}\theta, \quad \theta = [-\pi, \pi].$$

Similarly, for a multi-dimensional problem, we consider a continuous function $g(x_1, \cdots, x_s)$ defined on the range $\prod_{j=1}^{s}\left[\underline{x_j}, \overline{x_j}\right]$ with the degree $n_j$:

$$\begin{aligned} g(x_1, \cdots, x_s) \quad &\approx g_{F_n}(x_1, \cdots, x_s) \\ &= g_{F_n}(\theta_1, \cdots, \theta_s) \\ &= \sum_{i_1=0}^{2n_1} \cdots \sum_{i_s=0}^{2n_s} \left(\tfrac{1}{2}\right)^s g_{i_1, \cdots, i_s} T_{i_1, \cdots, i_s}(x_1, \cdots, x_s) \\ &= \sum_{i_1=0}^{2n_1} \cdots \sum_{i_s=0}^{2n_s} \left(\tfrac{1}{2}\right)^s g_{i_1, \cdots, i_s} T_{i_1, \cdots, i_s}(\theta_1, \cdots, \theta_s) \end{aligned} \tag{17}$$

where

$$\begin{aligned} g_{i_1, \cdots i_s} \quad &\approx \left(\prod_{i=1}^{s}\frac{2}{m_i}\right)\sum_{n_1=1}^{m_1} \cdots \sum_{n_s=1}^{m_s} g(x_{n_1}, \cdots, x_{n_s})T_{i_1, \cdots, i_s}(x_{n_1}, \cdots, x_{n_s}) \\ &= \left(\prod_{i=1}^{s}\frac{2}{m_i}\right)\sum_{n_1=1}^{m_1} \cdots \sum_{n_s=1}^{m_s} g(\theta_{n_1}, \cdots, \theta_{n_s})T_{i_1, \cdots, i_s}(\theta_{n_1}, \cdots, \theta_{n_s}) \end{aligned}$$

$$
\begin{aligned}
T_{i_1,i_2,\cdots i_s}(x_1, x_2, \cdots, x_s) &= T_{i_1,i_2,\cdots,i_s}(\theta_1, \theta_2, \cdots, \theta_s) \\
&= T_{i_1}(x_1) T_{i_2}(x_2) \cdots T_{is}(x_s) \\
&= T_{i_1}(\theta_1) T_{i_2}(\theta_2) \cdots T_{is}(\theta_s)
\end{aligned}
$$

where $m_i(i = 1, 2, \cdots, s)$ is the number of interpolation points for the corresponding variable.

Based on the theory of Fourier series polynomials approximation in the previous subsection, we can construct a Fourier series-based polynomials surrogate model:

$$
g(x_1, \cdots, x_s) \approx \sum_{i_1=0}^{2n_1} \cdots \sum_{i_s=0}^{2n_s} \left(\frac{1}{2}\right)^s g_{i_1,\cdots,i_s} T_{i_1,\cdots,i_s}(x_1, \cdots, x_s) = \boldsymbol{\beta}^T \boldsymbol{\alpha} \tag{18}
$$

where

$$
\boldsymbol{\beta} = [\beta_1, \cdots, \beta_m]^T = (1/2)^s [g_{0,\cdots,0}, \cdots, g_{i_1,\cdots,i_s}, \cdots, g_{2n_1,\cdots,2n_s}]^T
$$

$$
\boldsymbol{\alpha} = [\alpha_1, \cdots, \alpha_m]^T = [T_{0,\cdots,0}(\boldsymbol{x}), \cdots, T_{i_1,\cdots,i_s}(\boldsymbol{x}), \cdots, T_{2n_1,\cdots,2n_s}(\boldsymbol{x})]^T
$$

then

$$
\begin{bmatrix}
\alpha_1(\boldsymbol{x}_1) & \cdots & \alpha_m(\boldsymbol{x}_1) \\
\vdots & \ddots & \vdots \\
\alpha_1(\boldsymbol{x}_p) & \cdots & \alpha_m(\boldsymbol{x}_p)
\end{bmatrix}
\begin{bmatrix}
\beta_1 \\
\vdots \\
\beta_m
\end{bmatrix}
=
\begin{bmatrix}
g(\boldsymbol{x}_1) \\
\vdots \\
g(\boldsymbol{x}_p)
\end{bmatrix}
\tag{19}
$$

where $\boldsymbol{x}_1, \cdots, \boldsymbol{x}_p$ denotes the sample vector.

From Equation (19), the corresponding coefficient vector can be calculated numerically using the Gauss elimination method. Though the approximation accuracy of the higher order polynomial-based Fourier polynomial model is more suitable for strong non-linear problems than the traditional regression polynomial models (TRP) [14,21], especially for periodic problems, it is inappropriate for linear responses. Therefore, in order to make the Fourier series-based polynomial model suitable for linear response problems, the model should be augmented with a linear polynomial, given as [22]:

$$
g_a(\boldsymbol{x}) \approx \sum_{i_1=0}^{2n_1} \cdots \sum_{i_s=0}^{2n_s} \left(\frac{1}{2}\right)^s g_{i_1,\cdots,i_s} T_{i_1,\cdots,i_s}(\boldsymbol{x}) + \sum_{j=1}^{q} c_j f_j(\boldsymbol{x}), \tag{20}
$$

where $f(\boldsymbol{x})$ is a linear polynomial function, $q$ is the number of terms in the polynomial, and $c_j$ represent the unknown coefficients of the linear polynomial. It should be noted that the default order of the linear polynomial was set as 1 in this study.

Correspondingly, we can derive a group of linear equations with respect to the unknown expansion coefficients:

$$
\begin{bmatrix}
\alpha_1(\boldsymbol{x}_1) & \cdots & \alpha_m(\boldsymbol{x}_1) & f_1(\boldsymbol{x}_1) & \cdots & f_q(\boldsymbol{x}_1) \\
\vdots & \ddots & \vdots & \vdots & \ddots & \vdots \\
\alpha_1(\boldsymbol{x}_p) & \cdots & \alpha_m(\boldsymbol{x}_p) & f_1(\boldsymbol{x}_p) & \cdots & f_q(\boldsymbol{x}_p)
\end{bmatrix}
\left\{
\begin{matrix}
\boldsymbol{\beta} \\
\boldsymbol{c}
\end{matrix}
\right\}
=
\begin{bmatrix}
g(\boldsymbol{x}_1) \\
\vdots \\
g(\boldsymbol{x}_p)
\end{bmatrix}
\tag{21}
$$

By solving the unknown coefficients vector $\boldsymbol{\beta}$ and $\boldsymbol{c}$ in Equation (21), we can obtain the AFP surrogate model.

## 3. Design of Module

The conceptual missile design is an iterative process that requires several design iterations to achieve an optimal solution. Moreover, each iterative process needs to modify the corresponding configuration geometry parameters to obtain a new missile scheme for various requirements, and then carry out the structural numerical simulations for the new missile to evaluate against the requirements of strength and stiffness.

In order to realize the rapid and independent execution of the parametric modeling module and analysis module, we decided to use the UG/OPEN API secondary development tool to implement the development of the software with C/C++ language [23–25]. Note from Figure 3 that the software is divided into the parametric modeling module and analysis module, in which the analysis module is constituted by the process of model pretreatment and simulation calculation. For the transmission of information between modules, we developed a unified standard interface to extract and store features with UG/Open API tools, which can make the integration of various modules conveniently.

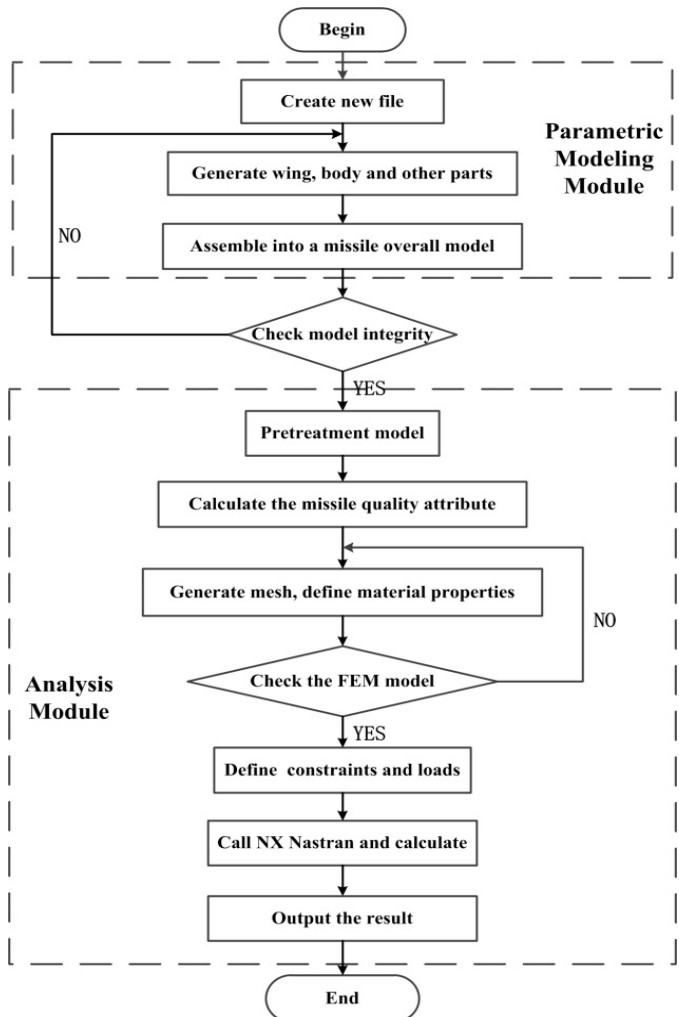

**Figure 3.** The flow chart of the software design.

### 3.1. Design of Parametric Modeling Module

The missile parametric modeling module is developed by the Open C programming interface based on UG/Open API tools. Because the basic form of Open C programming interface is simple and more convenient to compile when compared to Open C++. However, Open C++ can provide full access to its class hierarchy, allowing users to derive their classes by compiling new methods. Here, we choose the Open C programming interface to develop the parametric modeling module. The missile parametric modeling flow is shown in Figure 4.

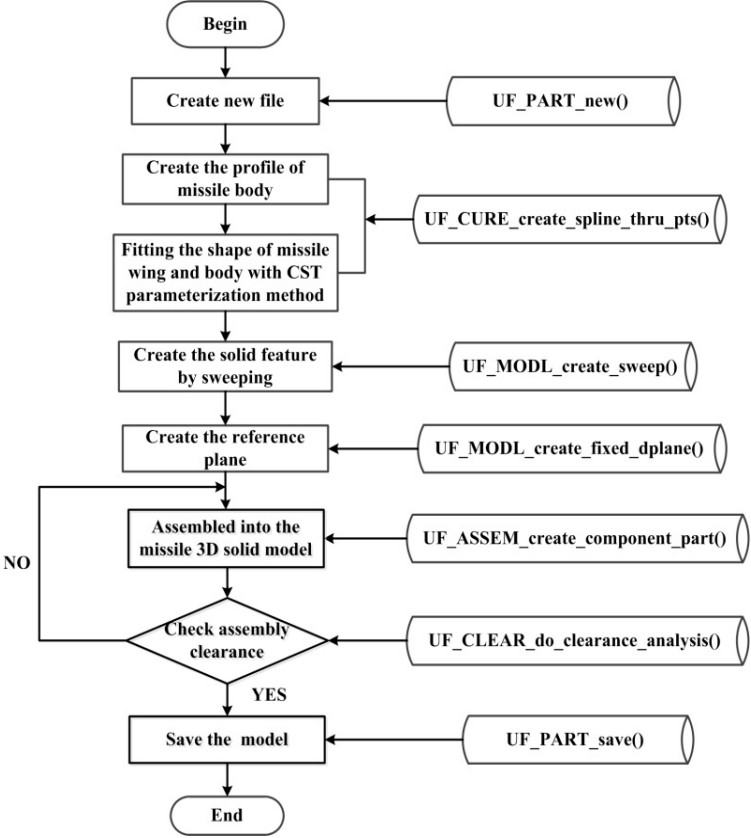

**Figure 4.** The flow chart of the parametric modeling module.

The missile parametric modeling module can be roughly divided into three parts: fitting curve, creating an entity model, and assembling the model. First, draw the contour line of the missile wing and projectile body by using the CST method. Subsequently, create the cross-section shape, and generate the solid model by sweeping. Then, assemble each component into the overall model. Finally, check assembly clearance to judge whether interference occurs between components in the assembly, the program will give a warning or error message when the check does not satisfy the requirements. Ultimately, the missile 3D entity model is established, and the model can be updated conveniently by modifying the expression values of geometric parameters.

The critical point of regular operation of the parametric modeling process is the feature extraction and storage, which involves the modification and invocation of the module. The feature information includes part name, dimension value, tolerance, and the coordinates on the reference coordinate system, and so on. Thus, the corresponding function is developed to traverse the entire assembly structure and store the information of all components, which can improve the versatility and convenience of the module.

## 3.2. Design of Analysis Module

The analysis module of the missile structure can be divided into two stages: model pretreatment and simulation calculation. The model pretreatment is the basis of simulation calculation, which guarantees the organic combination of the parametric modeling module and the analysis module.

The model pretreatment flow is shown in Figure 5. The specific implementation process can be summarized as follows: (1) open the model and read the feature information to prepare for the subsequent operation; (2) define the reference plane and trim the missile wing and projectile body along the reference plane to ensure a standard connection face; (3) divide the projectile body into many cabins and shell the model according to thickness requirements, which facilitate to assign various

material and thickness for various cabins; (4) create equivalent components at the corresponding location to simulate missile-borne equipment, such as the engine and fuel tank; (5) check the integrity of the model to prevent the appearance of detail parts when dividing the finite element mesh; and (6) calculate the quality attributes of the refined model and output the result.

In the stage of model pretreatment, the critical point is the target feature identification of the missile model. First, traverse the entire model to read the stored feature information. Then, establish the list of tags for all feature parameters used during the process. Finally, identify each component by tags to execute the subsequent parametric model treatment and analysis.

When the model pretreatment is completed, the program will enter the structural parametric analysis stage automatically. The task of structural analysis is to judge whether the strength and stiffness of the designed missile satisfy the requirements under the actual working conditions. The flow chart of the structure analysis process is shown in Figure 5. Note from the figure that the process is a standard finite element analysis step, which includes generating mesh, assigning material property, imposing constraints, applying the load, and calling the solver to calculate.

In order to improve the versatility of the analysis module in meshing, we adopt a generic approach: first, mesh each component separately and then connect two separate components and their associated 2D or 3D meshes by establishing a multi-point constraints equation. The mesh connects method in this paper can simplify the matching of various grids between components of the missile, and solve the problem of loads transfer between different types of elements. Additionally, it is crucial to verify the correctness of the finite element model, and test for cases such as duplicate elements or unconnected/un-equivalenced groups of elements. Hence, the function of the Finite Element Model Check will be executed to provide complete information about the model and all its finite element components to guarantee both the potential accuracy of the model and the overall quality of the finite element mesh.

Moreover, the missile's load is divided into two parts. The first is the aerodynamic force, which distributes on the upper and lower skin of the missile. Here, we use the equivalent concentrated load to simulate the distributed load to simplify the computational model, and then the program will automatically search for nearby nodes in the corresponding location to choose the loading points. The second is the self-structure weight, which loads with inertia force. For the imposing of constraints, we use the inertia relief method to simulate unconstrained missile structures in a static analysis, which use the inertia mass of the structure to resist the applied loadings to assume that the structure is in a state of static equilibrium even though it is not constrained.

Heretofore, the general modeling and analysis module of the missile structure can be completed through the process mentioned above, which can significantly improve the design efficiency of the missile and shorten the design cycle.

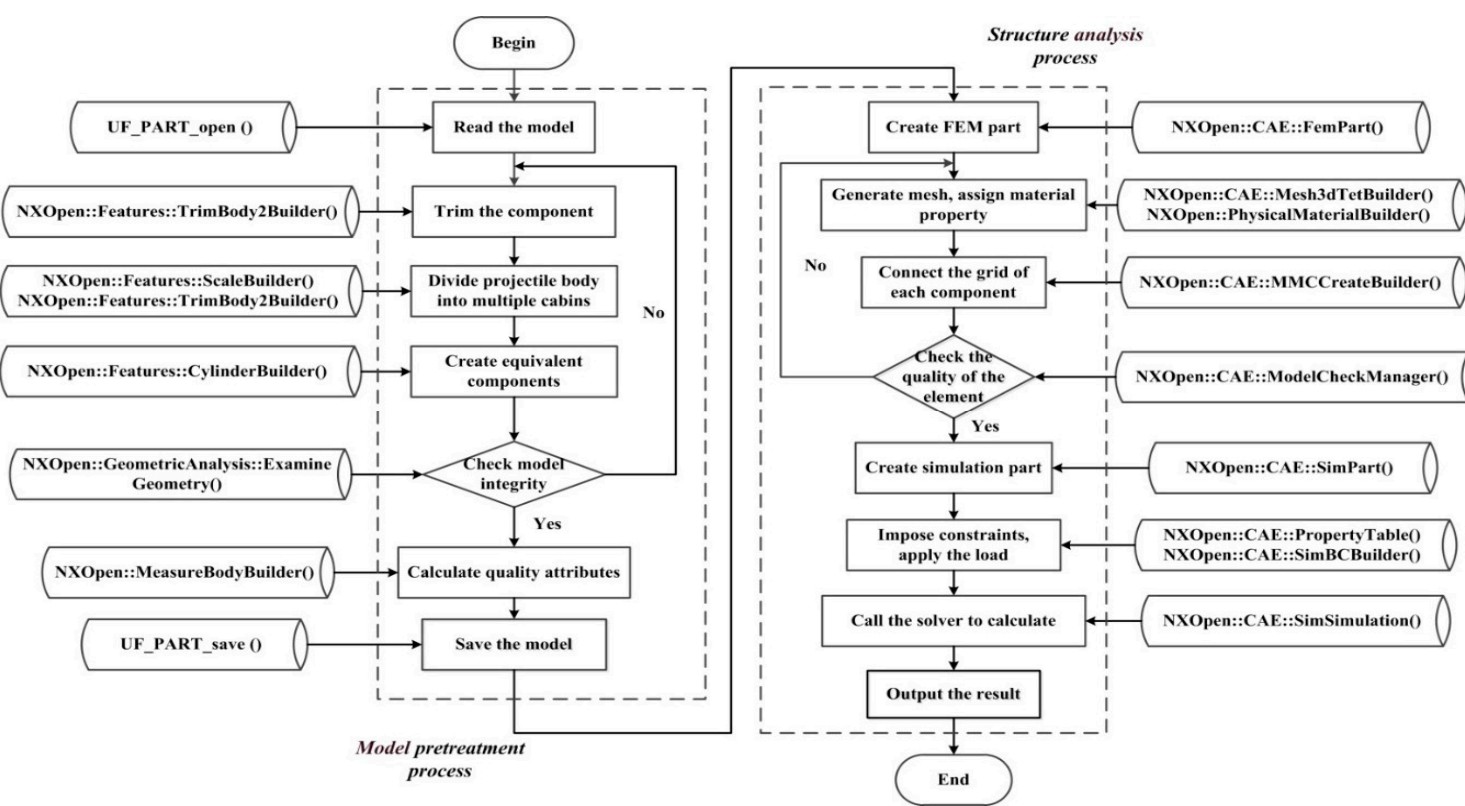

**Figure 5.** The flow chart of the analysis module.

## 4. Optimation Strategy

As mentioned in Section 3, the parametric modeling and analysis module of missile structure has been constructed, by which the original missile structure can be obtained and analyzed rapidly. However, due to epistemic factors, the original design is often not expected. In order to obtain a relatively optimal design rapidly, such as the light mass missile structure, the AFP surrogate model-based optimization strategy is designed in this section.

Moreover, the missile structure always involves a large number of parameters, which critically aggravate the calculation cost of the optimation process. Hence, it is necessary to determine which parameter exhibits the most effect on responses. In this study, the ANOVA technique is adopted to quantify the significance of parameters and subsequently utilized for parameter selection [26,27]:

$$
\begin{aligned}
Find \quad & \mathbf{x} = \{x_1, x_2, \cdots, x_n\} \\
\min \quad & f(\mathbf{x}) = \sum_{i=1}^{n} m_i = \sum_{i=1}^{n} \rho_i s_i x_i \\
s.t. \quad & g(\mathbf{x}) \le \sigma_{\max} \\
& x_i^l \le x_i \le x_i^u \qquad i = 1, 2, \cdots n
\end{aligned}
\tag{22}
$$

where $\mathbf{x}$ is the optimum thickness parameter vector of the system, $x_i^l$ and $x_i^u$ represent the available domain range of the $i$-th $\mathbf{x}$, and $n$ is the number of system characteristic parameters; for the objective function $f(\mathbf{x})$, different weight factors can be assigned to different output features, however, to simplify the analyzing process, we adopted the same weight factor for each part; $\rho_i$ and $s_i$ are the density and surface area of $i$-th part; $g(\mathbf{x})$ represents the stress function corresponding to the thickness parameter; and $\sigma_{\max}$ is the maximum allowable stress value.

Specifically, the flow chart of the missile structure optimation strategy is shown in Figure 6. It should be pointed out that the orthogonal experimental design method is adopted to obtain the sample for the execution of ANOVA. For the constructed AFP surrogate model, the accuracy is verified by solving N times with new randomly selected sampled points in the design space, and outputs of the AFP are compared with the corresponding results from FEM analysis software. Additionally, "end condition" is given as the form $f(\mathbf{x}) < \varepsilon$, where $\varepsilon$ is the convergence factor, and the "end condition" is considered to be met when the objective function is smaller than the convergence factor. Meanwhile, the judgment of "condition met" is an effective way to guarantee the regular operation of the optimation process.

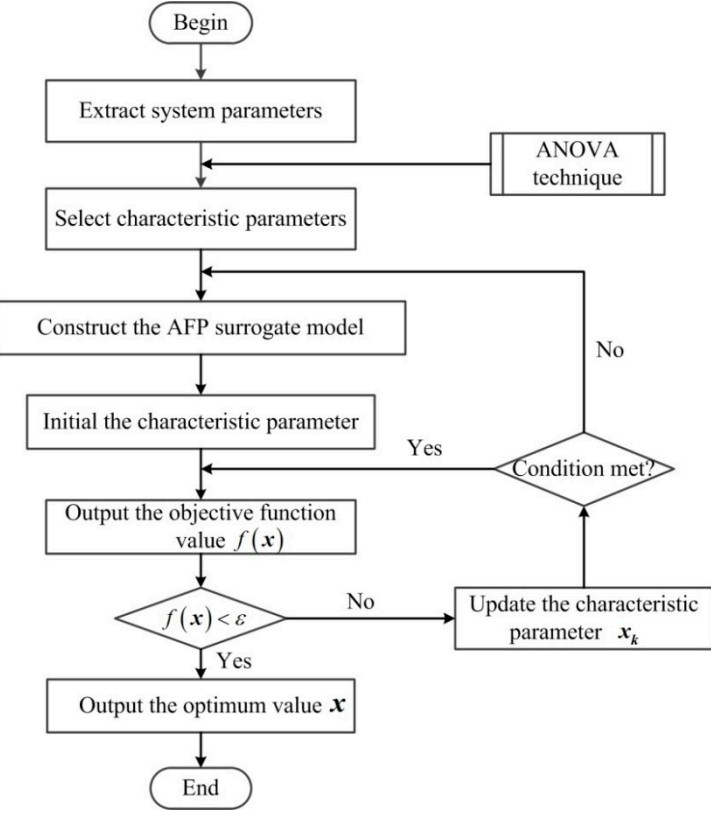

**Figure 6.** The flow chart of the optimation strategy.

## 5. Validation Examples

In principle, the parametric modeling module and structure analysis module developed in this paper can realize the process of rapid modeling and structural strength analysis of the new missile, and users can obtain the optimum design scheme of the missile with great value in engineering by modifying the various control parameters as needed. Therefore, several examples are presented below to verify the feasibility and versatility of the software.

### 5.1. Parametric Modeling

The missile model is a complex structure that includes plenty of geometry parameters, such as nose fineness, diameter, length, wing geometry/size, stabilizer geometry/size, and flight control geometry/size, among others. Some of these geometry parameters are listed in Table 1, for use in the subsequent parametric modeling process.

**Table 1.** Part of the control data of parametric modeling.

| Part | Parameter Names | Variables | Part | Parameter Names | Variables |
|---|---|---|---|---|---|
| | Total Length | Len | | Wingroot Shape | Swroot[ ] |
| | Nose Fineness Ratio | Rnd | | Wingtip Shape | Swtip[ ] |
| | Nose Length | Lnose | | Sweep Angle | Awsweep |
| Body | Boattail Length | Ltail | Wing | Wingspan | Lwspan |
| | Nose Shape | Snose[ ] | | Wing Position | Pwing[ ] |
| | Projectile Body Shape | Spb_1[ ] | | Installation Angle | Ains |
| | Boattail Shape | Stail[ ] | | Number of Wings | Nass |

Note from Figure 7, different missile models can be generated automatically just by modifying the various control parameters stored in the external text files. By contrast with the real missile [28], the validity and versatility of the parametric mode ling module can be verified.

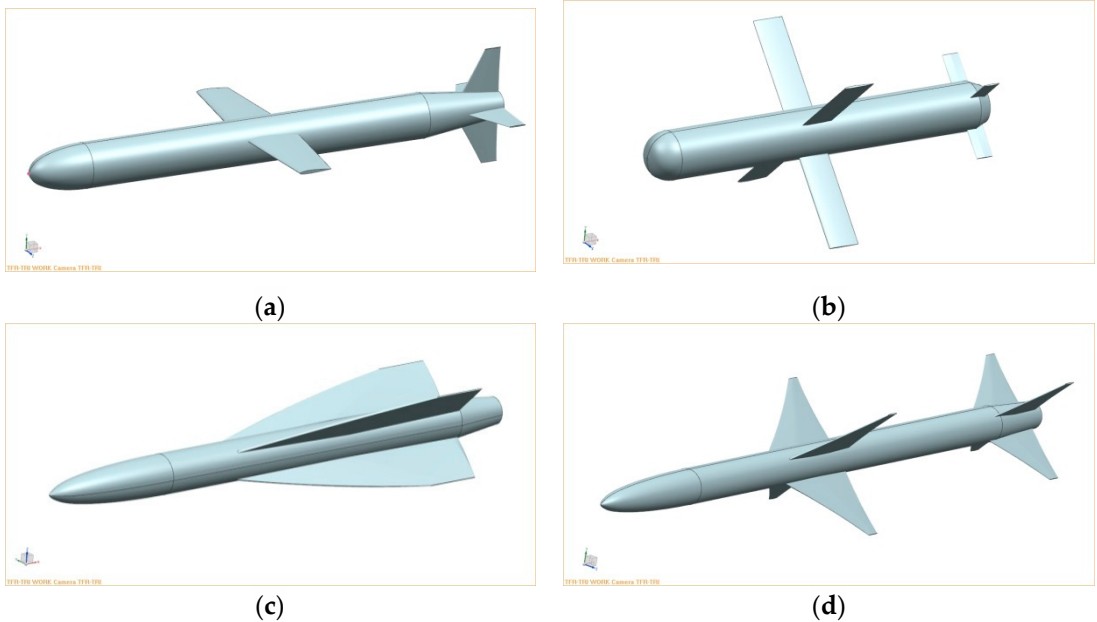

(a)

(b)

(c)

(d)

**Figure 7.** Different missile models:(**a**) "Tomahawk" missile model; (**b**) "PAM" missile model; (**c**) "MIM23" missile model; and (**d**) "AIM7" missile model.

### 5.2. Parametric Structure Analysis

In this section, we randomly select the "Tomahawk" missile as an example to verify the effectiveness of the subsequent parametric structure analysis and optimization modules. Because the analysis and optimization process of the other missile models is the same, in order to decrease the repeatability of this content, we will not repeat them here.

Similar to the Table 1, some necessary control parameters are listed in Table 2.

**Table 2.** Part of the control data of the parametric analysis.

| Part | Parameter | Variables | Part | Parameter | Variables |
|---|---|---|---|---|---|
|  | Number | Num_C |  | Type | Type |
|  | position | Pos_C[ ] |  | Size | Siz_W |
| Cabin | thickness | Thick | Wing | Material property | Mat_W[ ] |
|  | length | Len |  | Number | Num_W |
|  | Material | Mat_E[ ] |  | Position | Pos_L[ ] |
| Components | Position | Pos_E[ ] | Loads | Magnitude | Mag |
|  | Size | Size_E[ ] |  | Direction | Dir[ ] |

Since there is no detailed data of the Tomahawk missile, we assign a set of imaginary data to the control parameters, for example, cabin geometry/size, equivalent components geometry/size, material property, boundary constraints, and load conditions.

Figure 8 presents the pretreatment results of the validation case where the cabin is divided, and the equivalent components are assembled as required. Additionally, some detail parts of the model are processed to avoid the appearance of shape failure elements in meshing. Eventually, the quality attributes of the fine model have been calculated and output the result, which contains the model weight, volume, centroid coordinates, moment of inertia, and product of inertia.

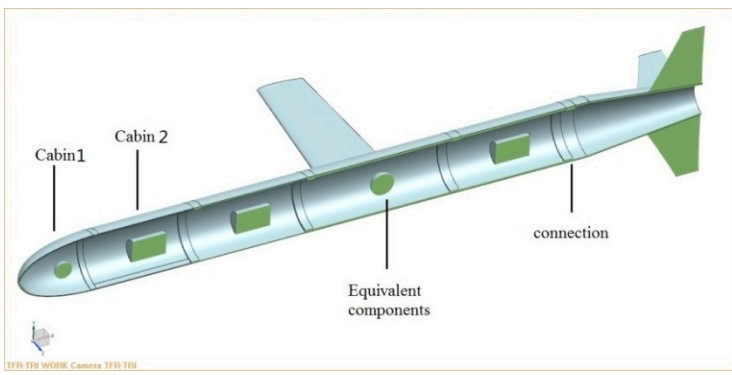

**Figure 8.** Missile model pretreatment.

Note from Figure 9 that the operation of mesh generation and loads applying is executed typically, which verifies the validity of model pretreatment and finite element modeling.

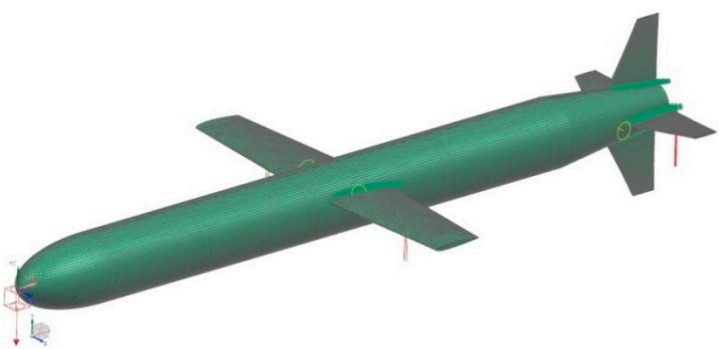

**Figure 9.** Mesh generation and loads applying.

After passing the Finite Element Model Check, the program will automatically call NX. Nastran solver to execute the static strength analysis of the missile. Figures 10 and 11 present the calculate results for displacement and stress, where the maximum displacement appears in the wingtip, and the maximum stress occurs at the wing root, which is similar to the result of the cantilever beam.

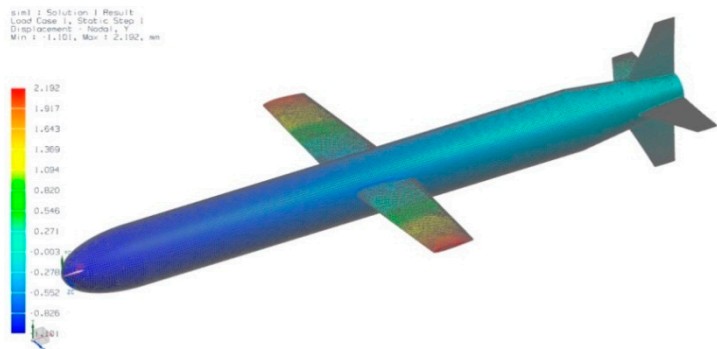

**Figure 10.** The displacement distribution chart of the missile.

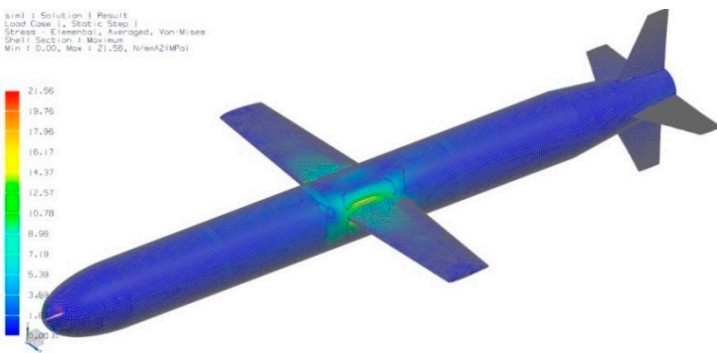

**Figure 11.** The stress distribution chart of the missile.

For the single modeling analysis process, we make a comparison of the traditional manual method with the parametric method developed in this paper. This comparison results are running under the computer hardware of Intel® Core™ i5 CPU and 8.00 GB RAM. The statistical time of the parametric method is from opening the external text files to completing all operations, which includes the time of modifying the control parameter as needed. Data in Table 3 is the statistical results of the authors' laboratory. The parametric modeling analysis method developed in this paper saves over 98% time comparing to the traditional manual method.

**Table 3.** Time comparison.

| Objects | Modeling Process | Analysis Process |
|---|---|---|
| Traditional manual | 1 h | 2 h |
| Parametric method | 20 s | 100 s |
| Saving time | 99.44% | 98.61% |

*5.3. Structure Optimation*

For the "Tomahawk" missile, we assumed the material of the body and wing is identical, and the nominal material properties of Young's modulus, the shear modulus, and the mass density were 69 GPa, 26 Gpa, and 2700 kg/m3, respectively. Moreover, the equivalent concentrated load is set as 104 N, and the maximum allowable stress value is set as 120 Mpa concerning the mechanical characteristics of the aluminum alloy. The complete set of potential input variables is listed in Table 4 based on design specifications and engineering experiences. The maximum stress values characteristics of the structure are applied as output variables.

**Table 4.** Input variables and description of the missile model.

| Index | Parameter | Description |
|---|---|---|
| 1 | $t_1$ | Thickness of cabin 1 |
| 2 | $t_2$ | Thickness of cabin 2 |
| 3 | $t_3$ | Thickness of cabin 3 |
| 4 | $t_4$ | Thickness of cabin 4 |
| 5 | $t_5$ | Thickness of cabin 5 |
| 6 | $t_6$ | Thickness of cabin 6 |
| 7 | $t_7$ | Thickness of wing |

The orthogonal experimental design method and ANOVA technique are performed to rank the importance of parameters on the features. As shown in Figure 12, the fourth and seventh parameters exhibit the most effect on structure maximum stress. Then, the thickness of cabin four and wing are chosen to be calibrated in the next procedure.

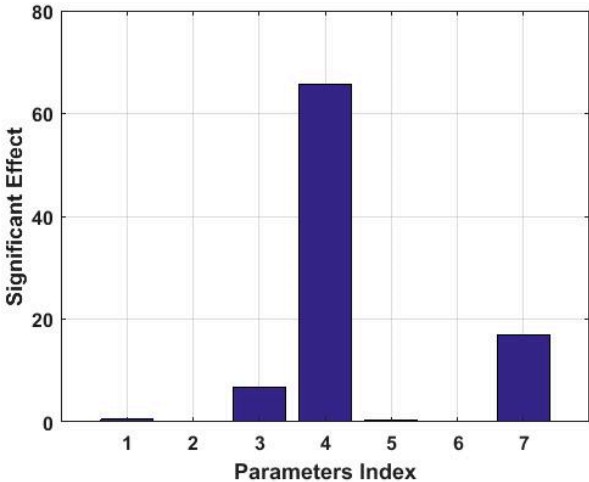

**Figure 12.** The stress distribution chart of the missile.

Afterwards, the optimization procedure of the missile structure can be formulated as the following model:

$$
\begin{aligned}
Find \quad & \boldsymbol{x} = \{x_1, x_2\} \\
\min \quad f(\boldsymbol{x}) &= \sum_{i=1}^{2} m_i = \rho(s_1 x_1 + s_2 x_2) \\
s.t. \quad & g_1(\boldsymbol{x}) \leq 120 \\
& g_2(\boldsymbol{x}) \leq 120 \\
& 4 \leq x_1 \leq 10 \\
& 6 \leq x_2 \leq 10
\end{aligned}
\tag{23}
$$

Then, the initial AFP surrogate model could be constructed, which $g_1(x_1, x_2)$ and $g_2(x_1, x_2)$ representing the maximum stress of the junction and loading point, respectively, were shown in Figures 13 and 14.

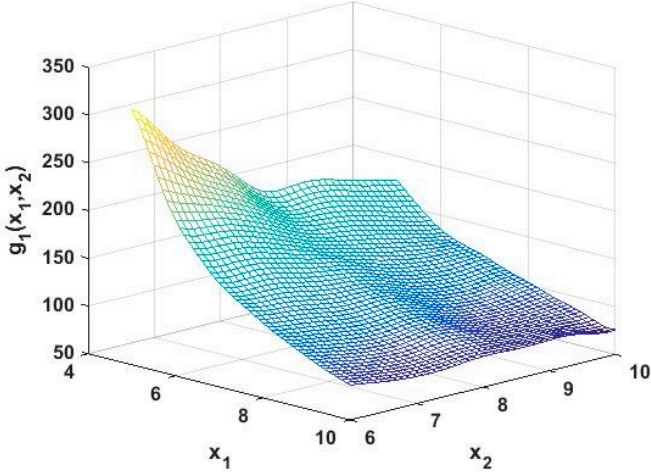

**Figure 13.** The augmented Fourier series-based polynomial (AFP) surrogate model of function $g_1(x_1, x_2)$.

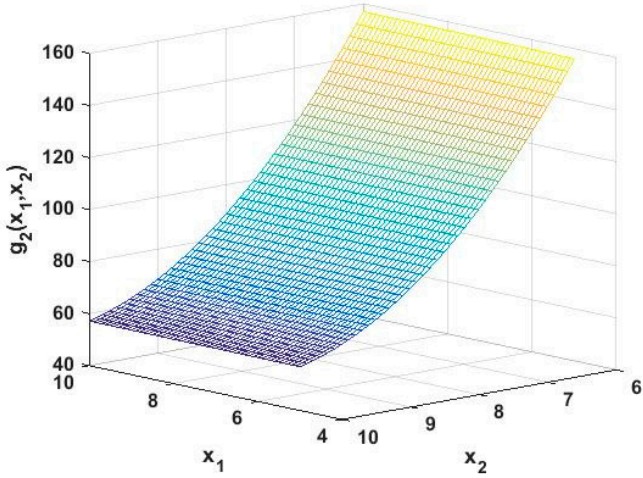

**Figure 14.** The AFP surrogate model of function $g_2(x_1, x_2)$.

After executing the procedure of structure optimization described in Section 4, the optimum value of the thickness is obtained; they are listed in Table 5. It can be noted from Figure 15 that the total mass of target components is considerably reduced from the initial 134.44 Kg to 100.58 Kg. Meanwhile, it can be seen that the convergence characteristic of the system optimization process is excellent, which demonstrates the effectiveness of the AFP surrogate model-based missile structure optimization method.

**Table 5.** Optimum results.

| Objects | Optimum Value |
|---|---|
| Thickness of cabin 4 (mm) | 7.96 |
| Thickness of wing (mm) | 6.94 |
| Weight (kg) | 100.58 |

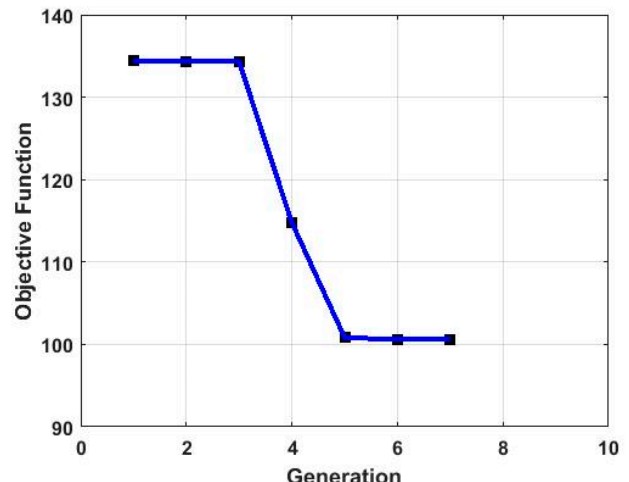

**Figure 15.** The convergence processes of the objective function.

## 6. Conclusions

In this study, a general integrated method for design, analysis and optimization of missile structure has been proposed for the rapid conceptual design of missile structure, by which the relatively optimal result can be obtained efficiently. Additionally, various missile examples are provided to verify the effectiveness of this method. From this study, the following conclusions can be drawn:

(1)    The CST parameterization methodology is derived, which can accurately model any airfoil in the entire design space with fewer variables;

(2)    The parametric modeling and analysis module of the missile is developed in C/C++ language on the basis of the CST parametric method and UG secondary development technology. Moreover, several various examples are presented to verify the validity and the versatility of this module;

(3)    By introducing a linear polynomial, a novel Augmented Fourier series-based polynomial surrogate model can be obtained. It can more accurately reflect the mathematical relationship between input and output data, which has the advantages of efficiency, transparency, and conceptual simplicity, especially for strong non-linear problems;

(4)    A novel surrogate model-based optimation strategy is designed to obtain a relatively light mass missile structure under the existing shape size, and satisfactory results are obtained when applied the structure optimation procedure to a missile example;

(5)    Compared with the traditional manual method, the parametric modeling and analysis method proposed in this paper can significantly improve the design efficiency of the missile, and shorten the design cycle, which provides a method with great value in engineering;

(6)    Subsequently, a comprehensive analysis platform of the missile can be established by using the parametric modeling module and the parametric analysis module developed in this paper, and other analysis modules, such as aerodynamic analysis module and stealth performance analysis module, which would provide a comprehensive method to realize the multidisciplinary optimization design of missiles.

**Author Contributions:** Conceptualization, J.G. (Jun Guo); Formal analysis, X.W. and J.Y.; Methodology, X.W. and J.G. (Jun Guo); Project administration, J.Y.; Software, X.W. and J.Y.; Validation, X.W.; Writing—original draft, X.W.; Writing—review & editing, J.G. (Jun Guo) and J.G. (Jian Guo)

**Funding:** This work was founded by the National Science Foundation of China (Grant No. 61374048).

**Conflicts of Interest:** The authors declare no conflict of interest.

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
