# Peer review of "A General Integrated Method for Design Analysis and Optimization of Missile Structure"

_algorithms, doi:10.3390/a12120257_

Round 1

Reviewer 1 Report

Authors have optimally designed and elaborated defense missile system.  The modelling and FEM analyses are well done.

Authors have decided to use CST parameterization methodology to represent wing and body shape of missile. In their one of the figures, they demonstrated the effectiveness of CST method. Shortening the design cycle of the missile structure definitely helps the intelligence sector. The novelty of this manuscript is construction of new augmented Fourier Series based polynomial. Authors have described the required steps for design of parametric modeling module and analysis module. Also, manuscript demonstrates several examples to validate the design scheme of structure which is highly significant to scientific community. It would be nice if the authors can compare parametric modeling and design to any of already existed missile. Overall, I would recommend its publication.

Author Response

Point 1: It would be nice if the authors can compare parametric modeling and design to any of already existed missile

Response 1: Thanks a lot for your recognition of our work and constructive suggestions on our manuscript. We have conducted a careful analysis and detailed investigation of your suggestions. Unfortunately, we find that the details of existing missiles are difficult to obtain. We can only simulate the condition of the entire missile based on some information. If we can get more detailed data in the future, we will supplement it according to your opinion. Thank you once again for your recognition of our work and constructive suggestions on our manuscript.

Reviewer 2 Report

The paper has an interesting purpose on the optimization design of missiles for the defense and war industries. The manuscript is well organized and the theoretical fundaments are clearly detailed. My major flaw is that it could be included the results not only for the Tomahawk missile, but also for the other missile types (e.g. PAM, MIM23, and AIM7). These results could support the conclusions and increase the impact of the research. It is also recommended an English revision of the paper.

Round 2

Reviewer 2 Report

The paper is now suitable for publication in the Journal.